# Efficient generation of entangled multiphoton graph states from a single atom

Philip Thomas[1✉], Leonardo Ruscio[1], Olivier Morin[1] & Gerhard Rempe[1]

The central technological appeal of quantum science resides in exploiting quantum effects, such as entanglement, for a variety of applications, including computing, communication and sensing[1]. The overarching challenge in these fields is to address, control and protect systems of many qubits against decoherence[2]. Against this backdrop, optical photons, naturally robust and easy to manipulate, represent ideal qubit carriers. However, the most successful technique so far for creating photonic entanglement[3] is inherently probabilistic and, therefore, subject to severe scalability limitations. Here we report the implementation of a deterministic protocol[4–6] for the creation of photonic entanglement with a single memory atom in a cavity[7]. We interleave controlled single-photon emissions with tailored atomic qubit rotations to efficiently grow Greenberger–Horne–Zeilinger (GHZ) states[8] of up to 14 photons and linear cluster states[9] of up to 12 photons with a fidelity lower bounded by 76(6)% and 56(4)%, respectively. Thanks to a source-to-detection efficiency of 43.18(7)% per photon, we measure these large states about once every minute, which is orders of magnitude faster than in any previous experiment[3,10–13]. In the future, this rate could be increased even further, the scheme could be extended to two atoms in a cavity[14,15] or several sources could be quantum mechanically coupled[16], to generate higher-dimensional cluster states[17]. Overcoming the limitations encountered by probabilistic schemes for photonic entanglement generation, our results may offer a way towards scalable measurement-based quantum computation[18,19] and communication[20,21].

Entanglement plays a crucial role in quantum information science. For multiqubit systems, many of the states considered, such as for entanglement purification, secret sharing, quantum error correction, as well as interferometric measurements, belong to the family of graph states[9]. Two prominent examples are GHZ and cluster states, which are central ingredients for various measurement-based quantum information protocols[19–21]. One-way quantum computing[18], for instance, represents a conceptually appealing alternative to its circuit-based counterpart. Instead of carrying out unitary quantum logic gates, computation relies on adaptive single-qubit measurements. This operational easing comes at the price that a multiqubit entangled resource state, a cluster state, needs to be prepared in advance.

Although multiqubit entanglement has been demonstrated on various platforms[3,22–26], only small-scale implementations of measurement-based quantum computing have been realized so far[10,27,28]. Among these platforms, optical photons stand out as qubit carriers, as these suffer negligible decoherence and benefit from crosstalk-free single-qubit addressability and measurement capabilities with off-the-shelf components. However, the most common sources for entangled photons are based on spontaneous parametric down-conversion (SPDC). This scheme is inherently probabilistic and thus makes scaling up to larger states an increasingly difficult challenge, even for a moderate number of qubits.

To address this issue, deterministic schemes have been proposed[4–6]. These use a single-spin memory qubit that mediates entanglement over a string of sequentially emitted photons. This approach is resource efficient as it permits the generation of in principle arbitrarily many entangled photons from a single device. First experiments along these lines have been performed with quantum dots[11,12] demonstrating entanglement of up to three and four qubits, respectively, in a linear cluster state. Low photon generation and collection efficiencies, a noisy semiconductor environment or the need for a probabilistic entangling gate were among the biggest obstacles for reaching higher photon numbers. Recent experiments with Rydberg superatoms[13,29] demonstrated GHZ states of up to six photons. Although the single-emitter strategy could in principle provide a stepping stone for photonic quantum computation, no implementation has demonstrated a performance that beats or even compares to the SPDC approach[3].

Here we produce large and high-fidelity photonic graph states of the GHZ and cluster type. Inspired by the proposals of refs. [4–6], which we adapt to our physical system, we use a cavity quantum electrodynamics platform as an efficient photon source[30–34] and, for the first time, surpass the state-of-the-art SPDC platform. Arbitrary single-qubit rotations between photon emissions allow for the flexible preparation of different types of states in a programmable fashion. We generate and detect GHZ states of up to 14 photons and linear cluster states of up to 12 photons with genuine multipartite entanglement. In principle, higher-dimensional cluster states can be created by coupling several sources[17], for example, by means of optically mediated controlled NOT gates of the kind demonstrated recently[16]. By virtue of this feature, so

[1]Max-Planck-Institut für Quantenoptik, Garching, Germany. ✉e-mail: philip.thomas@mpq.mpg.de

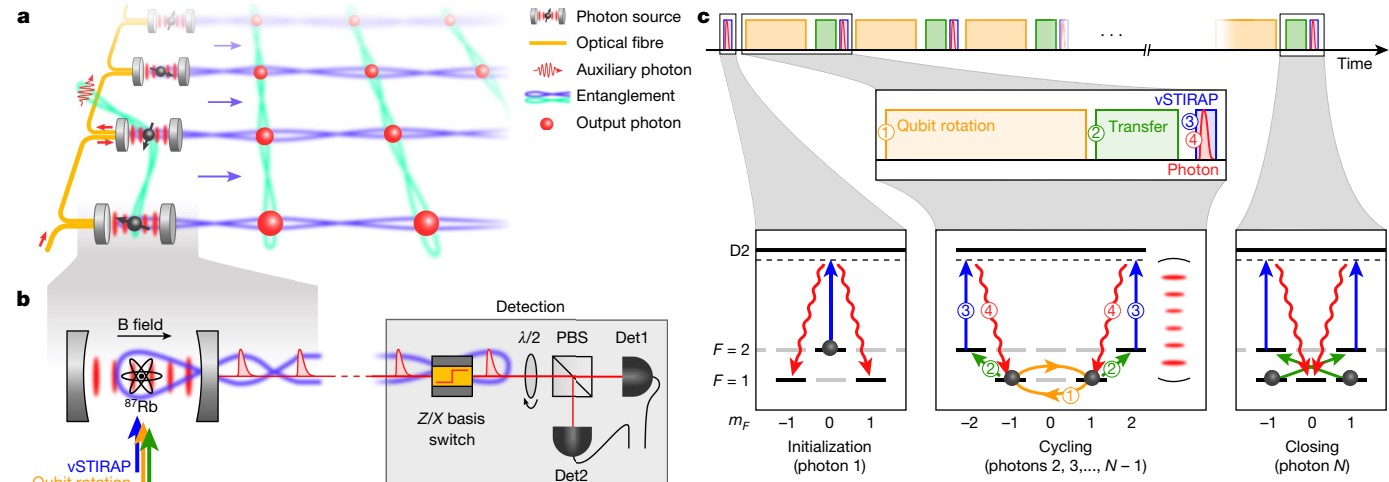

**Fig. 1 | Experimental setup and protocol. a**, Proposed one-way quantum computing hardware architecture. Each source produces a one-dimensional photonic cluster state while at the same time entanglement is distributed across the emitters by ancillary photons being successively reflected off each atom–cavity system[16]. Thus a two-dimensional fabric of photonic cluster states is woven from the individual one-dimensional chains, readily serving as a resource for measurement-based quantum computing. **b**, Experimental setup. A single $^{87}$Rb atom coupled to a high-finesse cavity emits a stream of entangled photons. Various lasers directed onto the atom from the side allow to control the photon emission and manipulate the atomic state to realize the desired protocol. The chain of photons is detected by a polarization-resolving

detection setup composed of a PBS and two detectors (Det1 and Det2). A fast polarization electro-optic modulator switches between two predefined settings such that each individual photon can be measured in either the $Z$ or the $X$ basis. A half-wave plate before the PBS rotates the detection basis from $X$ to $Y$ and arbitrary superpositions thereof. **c**, Protocol for cluster/GHZ state generation divided into three steps. Initialization of the atom and first photon emission, cycling process in which the atomic qubit is rotated by $\theta = 0$ or $\pi/2$ and a new entangled photon is emitted, and closing of the protocol by emission of a last photon while disentangling the atom from the generated string of photons.

far unique to the atomic cavity quantum electrodynamics platform, our technique supports modular extension towards scalable architectures for one-way quantum computation[18,19], as depicted in Fig. 1a.

## Experimental setup and protocol

Our apparatus is shown schematically in Fig. 1b. It consists of a single $^{87}$Rb atom at the centre of a high-finesse optical cavity. A magnetic bias field oriented parallel to the cavity direction defines the quantization axis and gives rise to a Zeeman splitting with Larmor frequency $\omega_L = 2\pi \times 100$ kHz. Several laser beams propagating perpendicular to the cavity allow for various manipulations, such as state preparation by optical pumping and coherent driving of Raman transitions between the hyperfine ground-state manifolds with energy selectivity provided by the Zeeman splitting. The cavity serves as an efficient light–matter interface for atom–photon entanglement[7] with an optical cooperativity of $C \approx 1.5$ (Methods). A vacuum-stimulated Raman adiabatic passage (vSTIRAP) enables the generation of photons with high indistinguishability stemming from accurate control over their temporal wave function[30]. Photons that are outcoupled from the cavity are analysed with a polarization-resolving detection setup mainly consisting of a polarizing beam splitter (PBS) and a pair of superconducting nanowire single-photon detectors. Furthermore, an electro-optic modulator is used for fast selection of the measurement basis by switching between the $Z$ basis (right and left circular polarization) and the $X$ basis (horizontal and vertical polarization). When set to the $X$ basis, a half-wave plate can optionally be placed to rotate the detection basis along the equator of the Bloch sphere.

The experimental protocol for generation of entangled photons in essence consists of a periodic sequence of photon generations interleaved with single-qubit rotations performed on the atom. The sequence is shown in Fig. 1c, including the corresponding processes in the atomic-level diagram. We first initialize the atom in the state $|F = 2, m_F = 0\rangle$ by optical pumping. Here we write the atomic state as

$|F, m_F\rangle$, in which $F$ denotes the total angular momentum and $m_F$ is its projection along the quantization axis. Then we apply a control pulse (1.5 µs), which induces the vSTIRAP process generating a photon (300 ns full width at half maximum) entangled in polarization with the atomic state. This process can be written as $|2, 0\rangle \rightarrow (|1, 1\rangle |R\rangle - |1, -1\rangle |L\rangle)/\sqrt{2}$, in which $|R/L\rangle$ denotes right/left circular polarization of the photon and $\{|1, 1\rangle, |1, -1\rangle\}$ serves as our atomic qubit basis. We then perform a single qubit rotation $R_\theta$ of angle $\theta$ (step 1) on the atom. For cluster states $\theta = \pi/2$, for GHZ states, no rotation is performed, that is, $\theta = 0$. Afterwards we transfer the qubit from $|1, \pm 1\rangle$ to $|2, \pm 2\rangle$ (step 2). Both steps 1 and 2 are realized by means of a series of Raman pulses with a 790-nm laser (Methods). Finally, we induce the vSTIRAP process (step 3) by applying a control pulse, which produces a photon (step 4) and takes the atom back to the states $|1, \pm 1\rangle$. Steps 2–4 can be summarized by writing $|1, \pm 1\rangle \rightarrow |1, \pm 1\rangle |R/L\rangle$. One photon production cycle consisting of steps 1–4 takes 200 µs (50 µs) for the cluster (GHZ) state sequence. It is repeated $N - 2$ times, each iteration adding another qubit to a growing chain of entangled photons. For the final photon however (closing), the atomic qubit is transferred from $|1, \pm 1\rangle$ to $|2, \mp 1\rangle$ (instead of $|2, \pm 2\rangle$) such that, in the subsequent emission process, the atom ends up in $|1, 0\rangle$, which readily disentangles it from the photonic state. We note that in the case of cluster states, initializing as well as disentangling the atom are not strictly necessary, as the same can be achieved by appropriate $Z$ basis measurements of the first and last photons[6]. In the case of GHZ states however, the protocol must be performed as described in Fig. 1c to obtain an $N$-photon state of the form $|GHZ_N\rangle = (|R\rangle^{\otimes N} + |L\rangle^{\otimes N})/\sqrt{2}$.

## GHZ states

We start the experiment by producing GHZ states. In contrast to cluster states, GHZ states are more sensitive to noise and require a higher level of control in their preparation process. Regardless, because their density matrix contains only four non-zero entries, it is much easier to

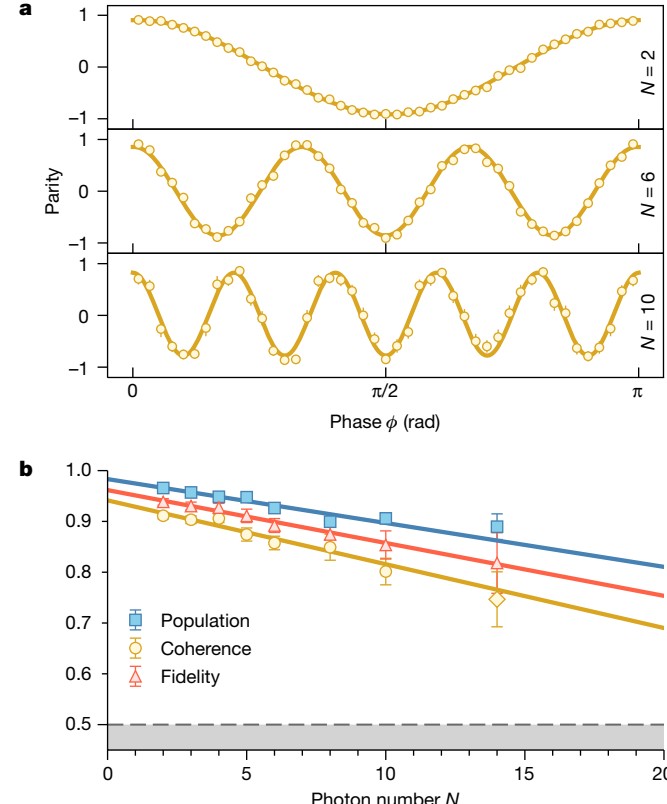

**Fig. 2 | GHZ states. a**, Parity measured in the basis $(|R\rangle \pm e^{i\phi} |L\rangle)/\sqrt{2}$ for photon numbers $N = 2$, 6 and 10 (see Extended Data Fig. 2 for the full dataset). The coherences $\mathcal{C}_N$ are derived from the visibility of the oscillations. **b**, Populations $\mathcal{P}_N$ (blue squares), coherences $\mathcal{C}_N$ (yellow circles) and extracted fidelities $\mathcal{F}_N$ (red triangles) with weighted linear fits as solid lines. The fidelity is calculated using the formula $\mathcal{F}_N = (\mathcal{P}_N + \mathcal{C}_N)/2$. For $N = 14$, the yellow diamond indicates that the coherence was derived from a single measurement setting with $\phi = 0$. The dashed grey line marks the classical threshold of 0.5. Error bars represent 1 standard deviation.

measure the fidelity $\mathcal{F}_N$ of an $N$-photon GHZ state[35] than for a cluster state, despite the large Hilbert space of dimension $2^N$. Therefore, the quantitative analysis of a multiphoton GHZ state, besides representing an interesting result by itself, provides a useful tool for benchmarking and gives insights into the inner dynamics of our system.

For estimation of the fidelity, it is sufficient to measure the non-zero elements on the diagonal and off-diagonal of the density matrix separately. The diagonal elements represent the populations $\mathcal{P}_N$ of the $|R\rangle^{\otimes N}$ and $|L\rangle^{\otimes N}$ components of the state and can be obtained by measuring all photons in the $Z$ basis. The corresponding experimental data, shown in Fig. 2b in blue, agree well with the ideal GHZ state, for which $\mathcal{P}_N = 1$, with only a weak dependence on $N$. To demonstrate that the states $|R\rangle^{\otimes N}$ and $|L\rangle^{\otimes N}$ are in a coherent superposition, we set the measurement basis to $(|R\rangle \pm e^{i\phi} |L\rangle)/\sqrt{2}$, in which $\phi \in [0, \pi]$, thus spanning the full equator of the Bloch sphere. This allows us to measure the characteristic parity oscillations, which behave as $\cos(N\phi)$ (Methods), see Fig. 2a. The coherences $\mathcal{C}_N$ of the density matrix are extracted from the visibility of the oscillations for all photon numbers up to $N = 10$. For 14 photons, the coincidence rate decreases notably owing to the finite photon production efficiency. To acquire enough data, we only measure the parity for $\phi = 0$, which is indicated by the yellow diamond in Fig. 2b. Eventually, the fidelity is calculated using the formula $\mathcal{F}_N = (\mathcal{P}_N + \mathcal{C}_N)/2$ and is shown in Fig. 2b in red. As only a single measurement setting was used for $\mathcal{C}_{14}$, we also provide a lower bound for the fidelity on the basis of an entanglement witness (Methods). With this,

we prove genuine 14-photon entanglement with a fidelity $\mathcal{F}_{14} \geq 76(6)\%$, surpassing the 50% threshold by more than 4 standard deviations. To the best of our knowledge, this is the largest entangled state of photons so far.

Within the measured range, we observe that the decay of $\mathcal{P}_N$, $\mathcal{C}_N$ and $\mathcal{F}_N$ as a function of photon number is well captured by a linear model with a slope of 0.86(9)%, 1.3(2)% and 1.04(9)% per photon, respectively. By extrapolation of this trend, we estimate that the fidelity will cross the 50% threshold at around 44 qubits. The remarkably slow decay in fidelity is particularly astonishing as we observe very little decoherence even when the sequence is deliberately chosen to exceed the intrinsic coherence time of the atomic qubit (about 1 ms). This behaviour is explained by a dynamical decoupling effect built into the protocol, which arises from the opposite signs of the Zeeman splitting in the two hyperfine ground-state manifolds. Hence, the qubit precession is reversed every time the atom is transferred from $|F = 1\rangle$ to $|F = 2\rangle$ or vice versa, which can be seen as two spin-echo pulses for every photon production cycle. Although this mechanism contributes to the high-visibility fringes seen in Fig. 2a, no extra effort is needed to exploit it (Methods). At present, we attribute the main source of infidelity to the vSTIRAP. This can be explained by the finite cooperativity that allows for unwanted paths in the emission process (Methods).

## Cluster states

The characterization of cluster states is more demanding, as the density matrix contains many non-zero elements. We therefore use the entanglement witness $\mathcal{W}$ proposed in ref. [36], which is based on the stabilizer formalism. A lower bound of the fidelity can be derived from $\mathcal{W}$, requiring only two local measurement settings $XZXZ\ldots$ and $ZXZX\ldots$ (Methods). Compared with quantum state tomography, this has the advantage of a tremendous reduction in experimental overhead but comes at the cost of a potentially substantial underestimation of the true state fidelity. Nonetheless, the experimental results shown in Fig. 3a exceed the 50% threshold for all measured points. Here, the data only includes events in which exactly $N$ photons are detected for a sequence of $N$ consecutive generation attempts. For the largest cluster state of 12 photons we find the fidelity to be lower bounded by 56(4)%. Comparing the results to the GHZ states in Fig. 2, we notice a much faster decay of the fidelity (3.6(2)% per photon). Besides the large number of Raman transfers in the protocol (five transfers per cycle, see Methods), we attribute this mainly to the lower bound, which—by construction—underestimates the fidelity. A tighter lower bound that was recently formulated[37] could provide a more accurate estimate of the fidelity in future experiments.

In addition to providing a lower bound for the fidelity, we now present the measured stabilizer operators, defined as $S_k = Z_{k-1} X_k Z_{k+1}$ (Fig. 3b). Here $k \in \{1, 2, \ldots, N\}$, $Z_0 = Z_{N+1} = \mathbb{1}$, and $X_k$ and $Z_k$ denote the respective Pauli matrices acting on the $k$th qubit. In this scenario, events in which three consecutive photons, $k - 1$, $k$ and $k + 1$, are detected in the appropriate basis contribute to the stabilizer $S_k$. In principle, arbitrarily many stabilizers could be measured by repeating the protocol for a corresponding number of cycles. Here, however, we terminate the sequence at $k = 15$. We find an average of $\langle S_1 \rangle = 96.13(9)\%$ and $\langle S_k \rangle = 92(1)\%$ for $k \geq 2$, indicating a large overlap of the generated state with the target linear cluster state.

## Coincidence rate

We emphasize that the ability of producing entanglement of up to 14 photons is based, on the one hand, on the excellent coherence properties of the atom and, on the other hand, on the large photon generation and detection efficiencies. The latter is crucial, as the success probability $p_s$ of detecting a coincidence of $N$ consecutive photons scales exponentially with the photon number, $p_s = \eta^N$. Here $\eta$ denotes the

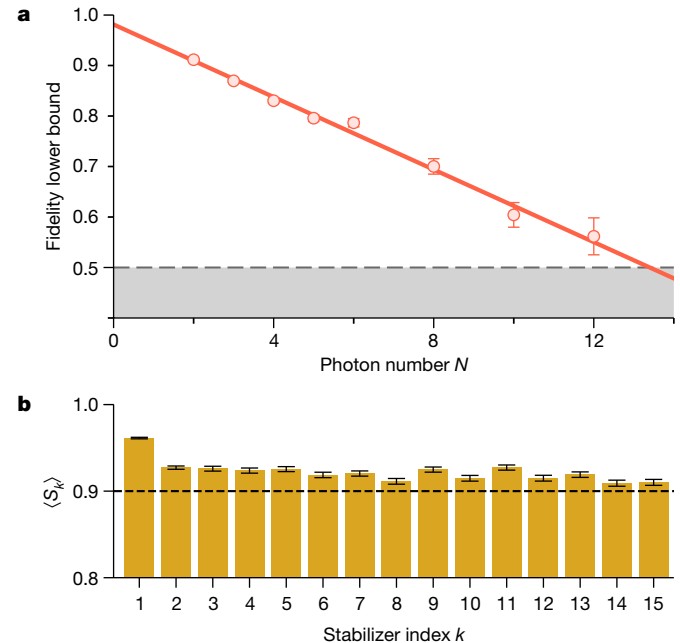

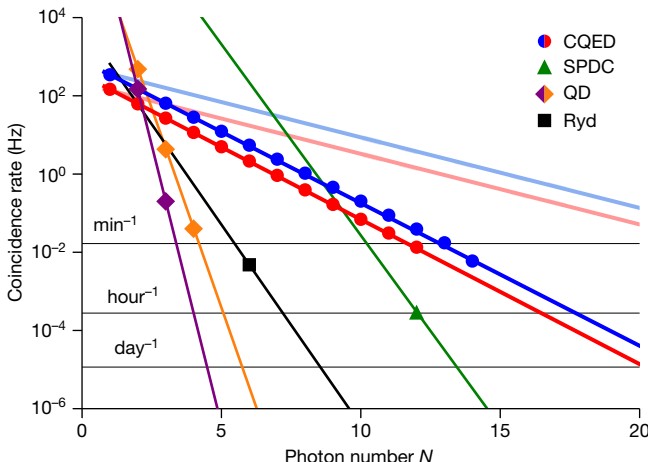

**Fig. 4 | Measured *N*-photon coincidence rate.** The data (blue for GHZ and red for cluster states) represent the number of coincidences divided by the total measurement time. From an exponential fit to the data, we extract a single-photon generation and detection probability of $\eta = 43.18(7)\%$. The light coloured lines represent the estimated coincidence rate assuming a loss-corrected efficiency of $\eta = 0.66$ (see also ref. [30]). As a comparison to our atomic cavity quantum electrodynamics (CQED) experiment, we plot equivalent rates for state-of-the-art SPDC[3], quantum dot (QD) in purple[11] and orange[12], and Rydberg-based[13] (Ryd) systems. Error bars are smaller than the markers.

**Fig. 3 | Cluster states. a,** Lower bound of the fidelity as provided by ref. [36] obtained from two measurement settings. For the measured data, the lower bound exceeds the threshold of 50% indicated by the grey dashed line. **b,** Measured stabilizers $S_k$ given by the expectation value $\langle Z_{k-1}X_kZ_{k+1}\rangle$ up to $k = 15$. All measured stabilizers are larger than 0.9, marked by the dashed line. Error bars represent 1 standard deviation.

probability to generate and detect a single photon for a given attempt. We can express $\eta$ as the product of the source efficiency $\eta_0$, that is, the probability of producing a photon at the output of the cavity, and the detection efficiency $\eta_d$. It is clear that a low efficiency $\eta \ll 1$ poses a great obstacle to achieving large photonic states within reasonable measurement times.

Figure 4 shows the raw rate of multiphoton coincidences as a function of photon number $N$ including post-selection and experimental duty cycle. The experimental sequence consists of 14 (12) consecutive photon generation attempts with all timing parameters identical to the GHZ (cluster) protocol and a new run starting every 1.1 ms (3.0 ms). The data shown (blue for GHZ and red for cluster states) are the coincidence count rates of events in which $N$ consecutive photons were detected starting from the first attempt. For instance, for the largest state of 14 photons, we recorded 151 coincidences in 7 h of experimental runtime, equivalent to roughly one event every 3 min. From an exponential fit to the data, we extract the overall single-photon generation and detection efficiency $\eta = 43.18(7)\%$. We estimate the intrinsic generation efficiency $\eta_0$ to be 66%, mainly limited by the cooperativity and the escape efficiency (see ref. [30]). Both can be optimized by higher-quality mirrors and a smaller cavity-mode volume. The detection efficiency of $\eta_d = 0.7$ captures all the remaining loss contributions, such as optical elements and detectors. These include free-space-to-fibre couplings (94% twice), propagation through optical fibre (97%), free-space optics (90%) and detector efficiency (90%). Correcting for the detection efficiency $\eta_d$, we infer an $N$-photon coincidence rate at the output of the cavity, as given by the light blue line in Fig. 4. This represents the limit of our system with the current parameters. As a comparison, we also show the rate of the best available SPDC system, as well as deterministic sources using single quantum dots or Rydberg-blockaded atomic ensembles. Although the repetition rate for these systems is typically many orders of magnitude higher than in our protocol, our system far outperforms previous implementations in terms of real-time coincidence count rate as well as efficiency scaling.

## Summary and outlook

To conclude, we have presented a scalable and freely programmable source of entangled photons, demonstrating—to our knowledge—the largest entangled states of optical photons to this day. It is deterministic in the sense that no probabilistic entangling gates are required. This gives us a clear scaling advantage over previous schemes. Moreover, the ability to perform arbitrary single-qubit rotations on the emitter provides the flexibility to grow graph states of different types.

At this stage, our system faces mostly technical limitations, such as optical losses, finite cooperativity and imperfect Raman pulses. Even modest improvements in these respects would put us within reach of loss and fault tolerance thresholds for quantum error correction[19,38–40]. Hence, a clear path towards one-way quantum computing architectures would be the generation of two-dimensional cluster states by entangling several photon sources[17]. For example, in a next step, two of our systems could be coupled through remote quantum logic gates[16] to produce $2 \times N$ 'ladder' cluster states. Alternatively, entangling operations such as gates or Bell-state measurements could be performed on two (or more) individual atoms as single emitters in the same cavity[14,15]. Similar strategies apply for the generation of tree graph states and one-way quantum repeaters[20,21]. The present work thus opens up a new road for photonic quantum computation and communication.

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

## Methods

### Experimental setup

The central component of the setup used in this work is a high-finesse optical cavity with a $^{87}$Rb atom trapped at its centre. The cavity consists of two highly reflective mirrors oriented parallel to each other at a distance of 500 μm with an optical mode waist of $w_0 = 30$ μm. The two mirrors have a transmitivity of $T_1 = 100$ ppm and $T_2 = 4$ ppm, giving rise to a finesse of $F \approx 60,000$, such that photons populating the cavity mode are outcoupled predominantly through the low-reflective side. The cavity is tuned to the atomic D2 line with a detuning of $\Delta_c = -150$ MHz with respect to the transition $|F = 1\rangle \leftrightarrow |F' = 1\rangle$. The combined system of the atom and cavity is best described in the framework of cavity quantum electrodynamics with parameters $(g, \kappa, \gamma) = 2\pi \times (c_{ge} \times 10.8, 2.7, 3.0)$ MHz, $g$ being the atom–cavity coupling strength for the relevant transition, $\kappa$ the decay rate of the cavity field and $\gamma$ the free-space atomic decay rate associated with the D2 transition of $^{87}$Rb. $c_{ge}$ is the Clebsch–Gordan coefficient between the relevant excited state ($|e\rangle$) coupling to the vSTIRAP control pulse and the final state ($|g\rangle$) of the photon production process. The above parameters put our system in the intermediate to strong coupling regime with a cooperativity parameter defined as $C = g^2/(2\kappa\gamma)$. Note that the specific value of $C$ depends on the transition path associated to a certain excited state. For example, for the emission from $|2, \pm 2\rangle$ as in the cycling step of the protocol, we have $|g\rangle = |F = 1, m_F = \pm 1\rangle$ and $|e\rangle = |F' = 2, m'_F = \pm 2\rangle$. Hence, we get $c_{ge} = \sqrt{1/4}$, giving $C = 1.8$.

### Atom loading

Atoms are transferred from a magneto-optical trap to the centre of the cavity, in which they are trapped by a two-dimensional optical lattice composed of two standing-wave potentials, one at 772 nm oriented along the cavity axis and one at 1,064 nm propagating perpendicular to the cavity axis. An electron-multiplying charge coupled device camera detects the atomic fluorescence, which is collected by means of a high-numerical-aperture objective. A single atom is prepared quasi-deterministically by removing any excess atom with a resonant laser beam steered onto the atom by means of an acousto-optic deflector. The position of the atom is monitored during the experiment and controlled through appropriate feedback to the optical trapping potential.

### Experimental duty cycle and post-selection

Because the atoms have a finite lifetime in the dipole trap, they have to be reloaded regularly. The average trapping time depends markedly on the type of conducted experiment (that is, heating/cooling mechanisms). For our experiments, we achieved an average trapping time of roughly 20 s. The time required for loading and repositioning of the atom after occasional jumps to a different location reduces the experimental duty cycle. By counting the number of experimental runs carried out at a given repetition rate over a longer measurement interval, we evaluate the overall duty cycle to be close to 50%.

Once a camera image shows that the atom has moved away from the target position, the corresponding data to that image are discarded through post-selection processing. The same applies to images with more than one atom near the cavity centre.

Further post-selection is performed by processing the data collected by the single-photon detectors: an experimental run is considered successful when $N$ photons were detected in a row, each within predefined time windows (1 μs width in this work). Note that Fig. 4 shows the coincidence rate after applying post-selection.

### Protocol

The full experimental sequence including timings of the optical pulses is shown in Extended Data Fig. 1. As described in the main text, it mainly consists of a repeating sequence of single-qubit rotations and photon emissions (cycling), with further initialization and closing steps at the beginning and the end. The atom is initialized in the state $|2, 0\rangle$ by optical pumping (5 μs). A square-shaped control pulse (1.5 μs) produces the first photon, thus generating the atom–photon entangled state $|1, 1\rangle |R_1\rangle - |1, -1\rangle |L_1\rangle$ (up to normalization), in which the index '1' refers to the first photon. If no photon was detected, we immediately go back to the state preparation step and another photon attempt. We choose a maximum of seven attempts for the first photon to avoid excessive heating of the atom. After a successful first photon detection, we start the cycling stage with the single-qubit gate, which—for cluster states—consists of a $\pi/2$ rotation contained in three Raman manipulations. First, the population in $|1, 1\rangle$ is transferred to $|2, 0\rangle$ with a $\pi$ pulse, taking 53 μs. Then a $\pi/2$ pulse is applied to the transition $|1, -1\rangle \leftrightarrow |2, 0\rangle$, realizing the qubit rotation. Afterwards, the population in $|2, 0\rangle$ is transferred back to $|1, 1\rangle$ with another $\pi$ pulse. The operation described transforms the basis states as follows: $|1, 1\rangle \rightarrow |1, 1\rangle + |1, -1\rangle$ and $|1, -1\rangle \rightarrow -|1, 1\rangle + |1, -1\rangle$. The whole pulse sequence for the single-qubit gate takes 132.5 μs. For GHZ states, the required rotation angle is $\theta = 0$, which means that the qubit rotation can be skipped entirely. To produce the next photon, we transfer the population from $|1, \pm 1\rangle$ to $|2, \pm 2\rangle$ by means of two sequential Raman $\pi$ pulses (790 nm), each taking 21 μs. We then apply a vSTIRAP control pulse, leading to a photon emission. The atom–photon–photon state then reads $(|1, 1\rangle |R_2\rangle + |1, -1\rangle |L_2\rangle) |R_1\rangle - (-|1, +1\rangle |R_2\rangle + |1, -1\rangle |L_2\rangle) |L_1\rangle$ for cluster states, that is, $\theta = \pi/2$. The index '2' now refers to the second photon. The cycling step is repeated as many times as desired. In the very last cycle, the closing step is performed. Here, following the qubit rotation, the atomic population is transferred from $|1, \pm 1\rangle$ to $|2, \mp 1\rangle$ instead of $|2, \pm 2\rangle$, which takes 55 μs. Thus the atom is disentangled in the subsequent photon emission. This step can be seen as an atom-to-photon state transfer, as the atomic qubit is mapped to the polarization state. After the last photon, we run a calibration sequence for actively stabilizing the optical power of the laser pulses. Finally, the atom is laser-cooled for several hundred microseconds. The length of a full period of the experiment including calibration and cooling depends on the type of state produced and the number of photons $N$. It can be as short as 400 μs and as long as 3 ms.

### Raman manipulations

The Raman transitions shown as orange and green arrows in Extended Data Fig. 1 are performed with a 790-nm laser. The duration of these transitions make up the most part of the experimental sequence. In principle, choosing a higher Rabi frequency could drastically increase the repetition rate of the protocol but would lead to more crosstalk between the transitions, as they would start to overlap in frequency space. As a consequence, a compromise between repetition rate and high-fidelity Raman manipulations has to be found. For our choice of experimental parameters, we estimate the infidelity per single-qubit rotation to be smaller than 1%.

The Raman transfer in the closing step from $|F = 1\rangle$ to $|F = 2\rangle$ is realized with a 795-nm Raman laser close to the D1 line of rubidium. For this specific Raman transition, we cannot choose a large detuning because this would lead to a destructive interference owing to the Clebsch–Gordan coefficients. As a consequence, we have a chance of about 5% of spontaneous scattering, which reduces the fidelity. As mentioned in the main text, alternatively, the atom can also be disentangled from the photonic state by measuring the most recently generated photon in the $Z$ basis. Although this would slightly increase the fidelity, the rate would decrease, as the detection of an extra photon is required.

### Estimation of errors

For GHZ states, we observe a total error rate of about 1% per photon. We attribute most of the infidelity to spontaneous scattering during the photon production process, as the vSTIRAP control pulse couples to

the $F' = 3$ excited state of the D2 line. This opens a decay channel, which competes with the coherent emission of the photon. By post-selecting on early photon arrival, one can partly filter out events in which scattering has occurred (Extended Data Fig. 4). In the future, this could be eliminated by generating the photons on the D1 line, in which no $F' = 3$ state is present. This should greatly improve the error rate.

The same error mechanism applies in the case of cluster states. Furthermore, the single-qubit gate implemented with Raman lasers introduces errors, which we estimate to be smaller than 1%. These could be explained by finite frequency resolution, pulse intensity fluctuations, as well as drifts in optical alignment. For instance, increasing the Zeeman splitting would be a way to further optimize this process.

Minor sources of error include polarization alignment. For setting the polarization detection basis, we use a reference polarizer in front of the cavity and measure the polarization extinction to be on the order of 10,000:1. For switching the detection basis, we use a polarization electro-optic modulator (QUBIG PC3R-NIR) with a switching time of 5 ns. The extinction ratio is specified as >1,000:1, whereas we measured values of around 5,000:1.

The error rate for cluster states of 3.6% as given in the main text is presumably overestimated owing to the definition of the fidelity lower bound. Taking into account the error sources identified above, we estimate the true error rate to be smaller than 2%. With the suggested improvements, we expect a reduction well below 1% to be realistic.

**Generation efficiency**

The intrinsic source efficiency, that is, the probability of obtaining a photon at the output of the cavity, is given by

$$\eta_0 = \frac{2C}{2C+1} \times \eta_{\text{esc}}, \tag{1}$$

in which $C \approx 1.5$ is the cooperativity and $\eta_{\text{esc}} \approx 0.88$ denotes the escape efficiency, that is, the probability of a photon being outcoupled from the output port[30]. Note that the above formula is only valid in the case of a single excited state, whereas the efficiency becomes a function of the detuning, $\eta_0(\Delta)$, when several excited states are present.

The source efficiency could hence be improved by increasing both the cooperativity and the escape efficiency. As the two parameters are generally not independent, let us assume for simplicity that we reduce the waist of the cavity mode by a factor of 2. This increases the cooperativity by a factor of 4 without altering the escape efficiency. We would thereby improve the source efficiency from 66% to 81%.

Furthermore, the efficiency of the detection setup could be improved. For instance, by redesigning and optimization of the setup, one could replace a fibre-to-fibre coupling with a fibre splice, eliminate a free-space-to-fibre coupling and reduce the losses from optical surfaces. In this scenario, an improvement of the detection efficiency $\eta_d$ from 0.7 to 0.85 seems feasible. Given these realistic improvements, the combined source-to-detection efficiency $\eta$ would reach the mark of 2/3, an important threshold for linear optical quantum computation[40].

**GHZ state fidelity**

In the mathematical formalism of spin 1/2 particles, a GHZ state looks like

$$|\text{GHZ}_N\rangle = \frac{1}{\sqrt{2}}(|\uparrow\uparrow\uparrow\ldots\rangle + |\downarrow\downarrow\downarrow\ldots\rangle), \tag{2}$$

in which in the photonic case $|\uparrow\rangle/|\downarrow\rangle$ corresponds to $|R\rangle/|L\rangle$. For measuring the diagonal elements of the density matrix, that is, the populations $\mathcal{P}_N$ of the $|\uparrow\rangle^{\otimes N}$ and $|\downarrow\rangle^{\otimes N}$ components, it suffices to measure all particles in the $Z$ basis to obtain

$$\mathcal{P}_N = \left\langle (|\uparrow\rangle\langle\uparrow|)^{\otimes N} + (|\downarrow\rangle\langle\downarrow|)^{\otimes N} \right\rangle. \tag{3}$$

For the coherences, we introduce the parity operator[3,35]

$$\mathcal{M}_\phi = \begin{pmatrix} 0 & e^{-i\phi} \\ e^{i\phi} & 0 \end{pmatrix}^{\otimes N} \tag{4}$$

describing a measurement of all $N$ particles in the basis $(|\uparrow\rangle \pm e^{i\phi}|\downarrow\rangle)/\sqrt{2}$. Varying the parameter $\phi$ from 0 to $\pi$ corresponds to a continuous rotation of the measurement basis along the equator of the Bloch sphere. In the experiment, this is achieved by scanning the angle of a half-wave plate in front of the PBS in the detection setup. It is straightforward to show that the expectation value of $\mathcal{M}_\phi$ for the ideal GHZ state is

$$\langle\text{GHZ}_N| \mathcal{M}_\phi |\text{GHZ}_N\rangle = \cos(N\phi). \tag{5}$$

These characteristic parity oscillations are what can be seen in Fig. 2a. The amplitudes of the oscillations as obtained from a cosine fit are a measure for the coherences of the density matrix. The fidelity is then obtained from the formula

$$\mathcal{F}_N^{(\text{GHZ})} = (\mathcal{P}_N + \mathcal{C}_N)/2 \tag{6}$$

For the largest photon number of $N = 14$, we chose to measure an entanglement witness derived in ref. [36] to obtain a fidelity lower bound. The witness is based on the stabilizer formalism, the stabilizing operators for GHZ states being

$$S_1^{(\text{GHZ})} = X_1 \cdot X_2 \cdots X_N \tag{7}$$

$$S_{k\geq2}^{(\text{GHZ})} = Z_{k-1} \cdot Z_k, \tag{8}$$

in which $k = 1, 2, \ldots, N$ and $Z_k$ and $X_k$ are the Pauli matrices acting on the $k$th qubit. With this, the fidelity is lower bounded by

$$\mathcal{F}_N^{(\text{GHZ})} \geq \frac{1 + S_1^{(\text{GHZ})}}{2} + \prod_{k\geq2} \frac{1 + S_k^{(\text{GHZ})}}{2} - 1. \tag{9}$$

**Witnessing cluster states entanglement**

A lower bound for the fidelity can be derived in a similar fashion for one-dimensional cluster states[36]. With the set of stabilizers $S_k$ as defined in the main text, the bound is given by the inequality

$$\mathcal{F}_N^{(\text{C})} \geq \prod_{k \text{ even}} \frac{1 + S_k}{2} + \prod_{k \text{ odd}} \frac{1 + S_k}{2} - 1 \tag{10}$$

It is easy to verify by direct calculation that the terms for even and odd $k$ in equation (10) correspond to the local measurement settings $ZXZX\ldots$ and $XZXZ\ldots$, respectively. As an example, for a four-qubit linear cluster state, we have

$$\begin{aligned} \mathcal{F}_4^{(\text{C})} \geq &\frac{1}{4}(1 + Z_1 X_2 Z_3)(1 + Z_3 X_4) \\ &+ \frac{1}{4}(1 + X_1 Z_2)(1 + Z_2 X_3 Z_4) \\ &- 1. \end{aligned} \tag{11}$$

**Coherence and dynamical decoupling**

In the main text, we already highlighted that our system benefits from a built-in dynamical decoupling mechanism owing to the level structure

of the atomic hyperfine ground states. A measurement of the intrinsic coherence time of the atom can be seen in Extended Data Fig. 3a. Here we look at the overlap between two photons both emitted from the atom with a variable time delay. The first photon is measured in the linear basis ($|H\rangle/|V\rangle$), which projects the atom onto a superposition of the qubit states $|1, +1\rangle$ and $|1, -1\rangle$. The atomic state then precesses with twice the Larmor frequency. After a certain time $t$, the atomic qubit is read out by mapping it onto a photon, which is then measured in the same basis as the first photon. The fidelity, which we define as the projection of the second photon on the first, shows oscillations damped by noise, such as magnetic field fluctuations. After roughly 1.2 ms, the envelope of the oscillations crosses the classical threshold of 0.66, which defines the intrinsic coherence time of the atomic qubit. For the GHZ sequence however, we observe that the effect of decoherence is intrinsically reduced. We can show this by artificially extending the length of the sequence to 1.25 ms for a six-photon GHZ state. In this case, every photon production cycle takes 300 µs. The ratio of time that the qubit resides in $|F = 1\rangle$ and $|F = 2\rangle$ can then be varied by scanning the delay $\tau$ between the hyperfine transfer from $|1, \pm1\rangle$ to $|2, \pm2\rangle$ and the vSTIRAP control pulse, as illustrated in Extended Data Fig. 3b. For different values of $\tau$, we record the parity oscillations similar to Fig. 2a and infer the visibility. From the measured data, we can see a clear dependence of the visibility as a function of $\tau$, with a rephasing appearing at around 85 µs. The maximum value is roughly equal to the six-photon coherence shown in Fig. 2 (shown as a dashed line for

reference), for which the sequence length was only 250 µs. This is strong evidence that a large part of the decoherence is mitigated as an inherent feature of the protocol.

## Data availability

The datasets generated and/or analysed during the current study are available at https://doi.org/10.5281/zenodo.6598546. Further information is available from the corresponding author on reasonable request.

**Acknowledgements** The authors thank A. Sørensen for valuable discussions. This work was supported by the Bundesministerium für Bildung und Forschung through the Verbund QR.X (16KISQ019), by the Deutsche Forschungsgemeinschaft under Germany's Excellence Strategy - EXC-2111 - 390814868 and by the European Union's Horizon 2020 research and innovation programme through the project Quantum Internet Alliance (QIA, grant agreement no. 820445).

**Author contributions** All authors contributed to the experiment, analysis of the results and writing of the manuscript.

**Funding** Open access funding provided by Max Planck Society.

**Competing interests** The authors declare no competing interests.

**Additional information**

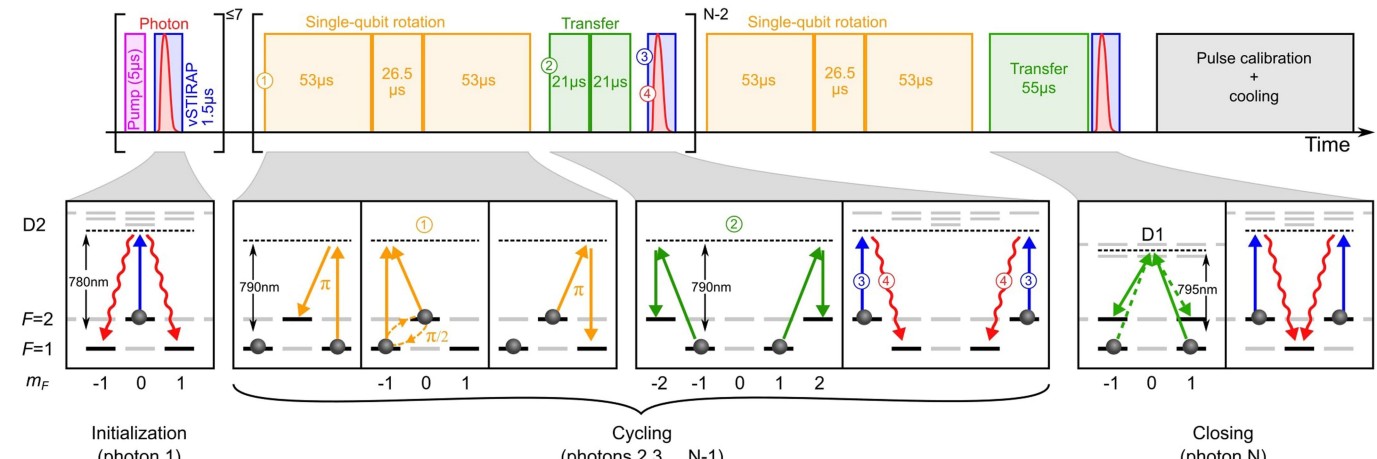

**Extended Data Fig. 1 | Detailed experimental sequence.** As in Fig. 1c, the sequence is divided into initialization, cycling and closing. After each run, we perform several hundreds of microseconds of active power stabilization of laser pulses, as well as atom cooling. The sequence shown takes up to 3 ms, depending on the number of photons and the type of photonic state (GHZ or cluster).

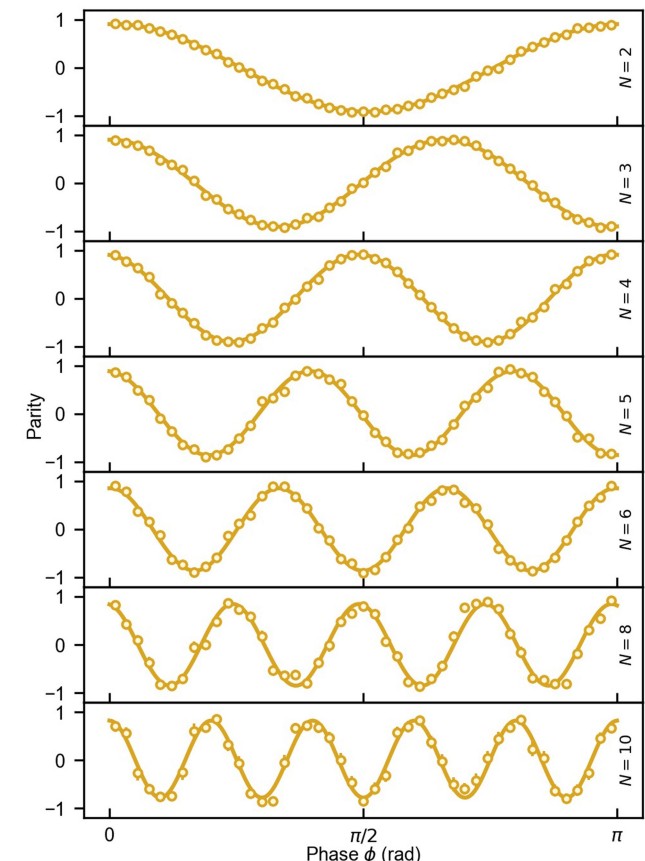

**Extended Data Fig. 2 | Parity oscillations.** Complete dataset for the GHZ coherence measurements for all measured photon numbers $N$.

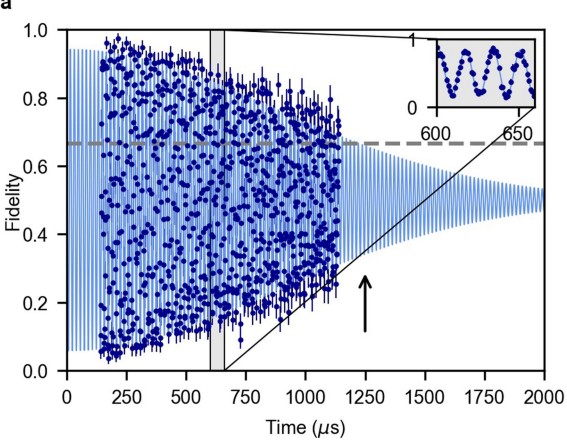

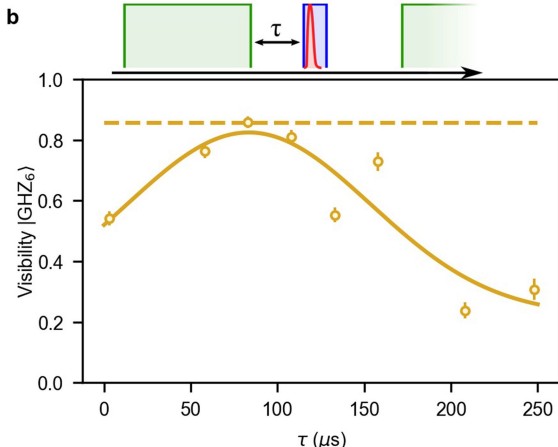

**Extended Data Fig. 3 | Coherence properties of the emitter. a**, Intrinsic memory coherence measured as the overlap between two photons emitted from the atom with a variable delay. The inset shows a zoom of the oscillations owing to the time evolution of the atomic qubit states. The arrow shows the sequence length for the data taken in **b**. **b**, Visibility of parity oscillations for a six-photon GHZ state as a function of temporal delay $\tau$ between the hyperfine remapping and the vSTIRAP control pulse. As a model is difficult to obtain for an unknown noise spectrum, a Gaussian fit to the data (solid line) provides a guide to the eye. A maximum of the visibility can be observed for around 85 µs.

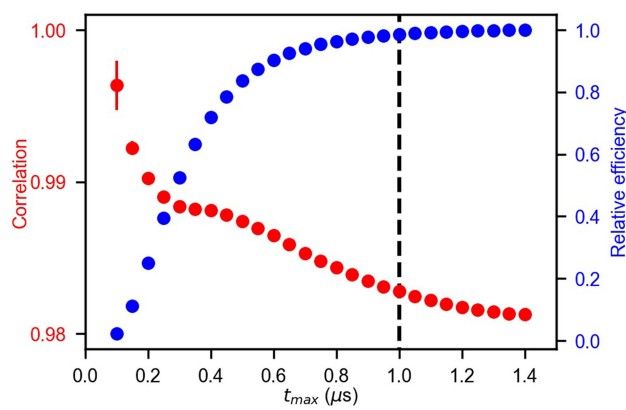

**Extended Data Fig. 4 | Infidelity induced by the vSTIRAP process.** Two photons are generated in subsequent cycles of the GHZ protocol and measured in the $R/L$ basis. Their correlation (red) is analysed as a function of maximum permitted arrival time $t_{max}$ with respect to the beginning of the emission process. The relative efficiency (blue) shows the number of counts detected up to $t_{max}$, as opposed to the full photonic wave packet. The correlation decreases as a function of $t_{max}$, which we attribute to spontaneous scattering events induced by the vSTIRAP control pulse. The dashed line marks the value of $t_{max}$ used in this work.