## [Peer Review File · Nature]

Manuscript Title: Efficient generation of entangled multi-photon graph states from a single atom

Reviewer Comments & Author Rebuttals

Reviewer Reports on the Initial Version:

Referees' comments:

Referee #1 (Remarks to the Author):

Thomas and co-authors report on a major experimental achievement, generating a highly-entangled state of many (14) photons. They obtain this result employing their latest cavity-QED system, based on technology developed and perfected for many years in the Rempe lab and elsewhere. This cavity-QED system constitutes a high-fidelity interface between a single atom and propagating photons. While efficient single-photon generation and atom-photon entanglement have already been demonstrated in such systems, the work reported in this Letter is the first time these processes are utilized sequentially beyond the few-photon level.

Two types of multi-partite entangled (graph) states are demonstrated: a GHZ state and a one-dimensional cluster state. Both are highly desired in the developing field of photonic quantum technology, including for computation, communication, and sensing. The degradation of the state fidelity with the number of photons is on the order of a few percent, significant and yet impressively low. With that, and with high efficiency of photon generation and collection, the demonstrated operation surpasses previous achievements in any other platform in both photon number and generation rate.

Highly-entangled multi-photon states can be produced with linear-optics entangling operations, which are inherently probabilistic. This approach has been successful in recent years, getting to the order of 20 photons and taking advantage of solid-state photonic technology. However, its probabilistic nature poses a major limitation on the generation rate and on the ability to scale up. In contrast, entangling operations based on a coherent light-matter interface, such as a cavity-single-atom setup, can be deterministic in principle and are thus more natural for manipulating many photons. To surpass the linear-optics performance, the quality of these interfaces has had to improve over the last two decades. In this paper, in my opinion, it reaches the break-even point. The demonstration of generating a graph state with desired connectivity of that many photons using an atom-photon interface is a significant milestone for quantum optics, which will be of interest to the whole quantum community. The consequences, particularly the path towards generating two- and three-dimensional cluster states as a resource for fault-tolerant photonic quantum computation, are extremely important.

The reported data are of high quality and appear valid. The methodology, particularly the estimation of fidelity and its statistical analysis, is sound. The manuscript is well written and relatively accessible. I have a few remarks and suggestions, and I support the publication in Nature.

1) The decay of fidelity of 1%-4% per photon is among the impressive results of the work. I propose that the author devote a short discussion in the main text to quantitatively break down the physical or technical factors contributing to these values.

2) Moreover, despite that decay being remarkably slow, it is still high for the needs of fault-tolerant measurement-based quantum computation. While the Outlook section alludes to different possible improvements, can the authors provide quantitative estimations of these improvements and a foreseen trajectory?

3) Arguably, the prime parameter of a coherent light-matter interface is the optical cooperativity. The main text mentions the cooperativity but does not provide its value. In continuation with the previous two comments, it would be helpful to quantify the cooperativity, explain the role it plays in the generation fidelity, perhaps what currently limits it, and whether higher cooperativity is feasible and useful.

4) In the GHZ measurements (e.g., for the data shown in Fig. 2), does N stand for the number of cyclings or the number of detected photons? On a minor note, please consider adding to Fig. 1b the wave-plate that sets ϕ in the GHZ measurements.

5) Line 47: An explicit citation to an SPDC system (e.g., Ref 3) can be provided at the end of the paragraph.

6) Line 79: The sentence (“The atom is hence in a superposition...”) is not strictly correct, as the atom is entangled with a photon.

7) Line 148: It can be helpful for the nonexperts to add a short comment on the implications of the measured values for the stabilizers.

Referee #2 (Remarks to the Author):

The manuscript reports on the largest entangled states of optical photons up to date. The experiment has been performed via an almost-deterministic protocol with one atom in a high-finesse optical cavity. The photons are generated via a memory protocol and entangling operations are performed via atomic-qubit rotations. The generated states are GHZ states and cluster states involving respectively up to 14 and to 12 photons.

The experiment is an important breakthrough, especially for the implementation of measurement-based quantum computing protocols in photonics. It indeed demonstrates the first implementation of single-photon cluster states without using probabilistic entangling gates, so opening the door to scalable photonics quantum computers.

I have nevertheless some comments about the reported claims that if addressed, without diminishing the importance of the presented results, would put the results into a clearer context.

The generation rate for GHZ states is reported as being faster than all the other recent implementations. In particular, in Figure 4, the coincidences rate versus photon-number of the GHZ protocol is compared with the ones of SPDC, Rydberg, and quantum-dot (QD) platforms. However, the reported results for QD platforms concern the generation of cluster states and not of GHZ states. As the generation rate of clusters and GHZ states in the manuscript are different, it would be fair to compare the coincidence rate of the QD experiments with rate of cluster states generation.

Also, quantum computing requires to go beyond linear cluster, by producing $n \times m$ grid cluster states. In the abstract, the authors predict modular extension via multiple atoms in a single cavity or by coupling several cavities, but given the cited articles (where 2 atoms in a cavity or 2 cavities are coupled) it would be more appropriate to promise $2 \times m$ - i. e. two dimensional - cluster states, which would be in any case an unprecedented result for single-photon cluster states.

On the same line: the coincidence rates scaling is driven by the overall generation and detection efficiency and in Figure 4 the light blue line indicates the rate corrected by the detection efficiency. Is there any room for fast improvement? i.e., the reported detection efficiencies (free-space-to-fiber couplings (94% twice), propagation through optical fiber (97%), free-space optics (90%) and detector efficiency (90%)) are already at the state of the art or they can be easily improved? Also, intrinsic generation efficiency can of course be improved via higher-quality mirrors and a smaller cavity mode volume, but this would require to build a new experimental setup. Those are parameters that probably have already been optimized according to other requirements, is this really a viable solution or it would degrade other figures of merit?

A minor point: the entangling procedure by repeating step 1-4 of the protocol is not easy to follow if the protocol proposed by Lindner and Rudolph in reference 6 is not already known to the reader. I suggest to give the explicit expression of the atom-photons states for at least two cycles in the Methods section.

Referee #3 (Remarks to the Author):

The manuscript by Thomas et al reports on the generation of 1D multi-photon entangled states using a single atom trapped in a cavity. Up to 14 photon GHZ states and 12 photon cluster states are realized making this the largest multi-photon entangled state generated to date and at a rate far beyond previous results. This is to be considered a major experimental achievement by realizing an on-demand entanglement protocol that has been a long-standing challenge in the field. The paper is very clearly and well written and the experimental realization is sound and convincing. In my opinion these achievements warrant publication of the work in Nature after thoroughly considering the following suggestions for improvements.

The weakest point of the work is potentially that it is not entirely clear what the next step of this research direction would be, i.e. what new opportunities does this work open up? In the introduction the authors state that the work "removes a long-standing obstacle towards scalable measurement-based quantum computation and communication" which is probably somewhat

overstated or maybe not sufficiently precisely elaborated in the text. The work demonstrates a way of creating 1D entanglement (which is not a sufficient resource for the mentioned application areas) and it is only hinted how to scale up to more advanced resource states by reflecting photons off the quantum nodes (Fig. 1a). Another approach would be to de-multiplex the 1D train of single photons and use quantum interference to synthesize more advanced states. However, this would require a high degree of indistinguishability of the emitted photons, which is not measured in the present experiment. Ideally the indistinguishability would be explicitly measured, since this is a key property of an entanglement source. I suggest the authors elaborate this point and give more precise indications of how to scale up the system and what potential applications that would be following.

Benchmarking with other platforms: the experiment is impressive and certainly the front-runner, which is clear from Fig. 4. However, from reading the manuscript one may get the incorrect impression that the single-atom approach has fundamental advantages compared to other emitters, which I don't think is appropriate. I think a more precise distinction would be between deterministic (Rydberg-based, quantum dots, SiV, or even superconducting qubits) and non-deterministic sources (SPDC). I would therefore suggest a more balanced description of the various deterministic sources including pros and cons. On a related note, I think the reference list is somewhat narrow and could be extended to better reflect the potential of the other deterministic platforms as well.

A few minor comments:

The sentence "By virtue of this feature, so far unique to the atomic CQED platform.." is not entirely clear. Why is this unique to atomic systems as opposed to other deterministic sources?

It would be interesting to have further details on the various contributions to infidelities in the experiment for instance by breaking down the protocol in the various steps and explain the error contributions (e.g. in Supplementary Information). That would help understand what are the critical steps for improving even further in the future. Also it would be instructive with a little more experimental details. How many actual coincidence events (14 fold) are detected and what was the experimental measurement time? What kind of post-selection is implemented and further elaboration of the role of the experimental duty-cycle could be given. What are the specs of the Z/X switch and how is the correlation detection carried out? In general an elaborated Supp. Info with such experimental details would be beneficial.

RESPONSE TO REFEREE #1

We thank the Referee for his/her thorough report and the valuable remarks that helped to improve our manuscript. We appreciate that our work is acknowledged as a “major experimental achievement” and a “significant milestone in quantum optics”. We are glad that the Referee supports publication in Nature.

Here is our point-to-point reply:

Referee: “1) The decay of fidelity of 1%-4% per photon is among the impressive results of the work. I propose that the author devote a short discussion in the main text to quantitatively break down the physical or technical factors contributing to these values.

2) Moreover, despite that decay being remarkably slow, it is still high for the needs of fault-tolerant measurement-based quantum computation. While the Outlook section alludes to different possible improvements, can the authors provide quantitative estimations of these improvements and a foreseen trajectory?

3) Arguably, the prime parameter of a coherent light-matter interface is the optical cooperativity. The main text mentions the cooperativity but does not provide its value. In continuation with the previous two comments, it would be helpful to quantify the cooperativity, explain the role it plays in the generation fidelity, perhaps what currently limits it, and whether higher cooperativity is feasible and useful.”

As noted by the Referee, the three questions are related. Therefore, we answer all of them at once. Before doing so, we would like to emphasize that investigations of such small errors are not straightforward, mainly because large amounts of data are needed to understand the different contributions to the errors. Our conclusions presented below might thus be somewhat preliminary.

In the case of cluster states, the largest imperfection comes from the construction of the estimation that, by definition, gives only a lower bound for the fidelity. Clearly, this is not an intrinsic source of infidelity and would not have to be accounted for when aiming for fault-tolerance thresholds. However, at this point we cannot state a precise number for its relative contribution to the total infidelity.

For the remaining infidelity of both cluster and GHZ states, we attribute the main contribution to the vSTIRAP process. In the Methods we added a new figure where we plot the two-photon polarization-correlation as a function of the maximum arrival time with respect to the beginning of the vSTIRAP process. Effectively, reducing the maximum arrival time selects photons emitted at the beginning of the vSTIRAP process, thus reducing the probability of measuring a second (unwanted) photon emission. As can be seen from the figure, this increases the correlation. Obviously, the vSTIRAP process is the main limitation for the fidelity. We now list this and other error contributions in a new section of the Methods and extrapolate to less than 1% error we could reach with realistic improvements.

Concerning the optical cooperativity, an explicit number was not given in the main text as it varies depending on the specific photon generation process. As can be seen in Figure 1c, the vSTIRAP process is initialized from different m_F states throughout the protocol. Furthermore, multiple excited states may contribute to the process, to each of which a different

number for the cooperativity has to be assigned according to the specific Clebsch-Gordan coefficient. Nonetheless, these numbers are of the same order of magnitude and we now give an approximate value in the main text as well as a more detailed explanation in the Methods.

The relationship between the optical cooperativity and the qubit fidelity as asked in question 3) is indeed relevant: in a simple picture, a finite cooperativity permits the atom to be excited to $F' = 3$ followed by a spontaneous decay back to $F = 2$, potentially followed by a second emission, and so on. This introduces errors. Although theoretical models have been developed describing the relationship between the cooperativity and the generation efficiency, we are not aware of any quantitative description that is suitable to explain the effect on the fidelity of polarization qubits for our system. However, we are fairly certain that a higher cooperativity would be beneficial as spontaneous processes such as scattering would be suppressed. Besides improving the cooperativity with a smaller cavity, we believe that working on the D1 line (where no $F' = 3$ excited state is present) should improve the fidelity.

Referee: "4) In the GHZ measurements (e.g., for the data shown in Fig. 2), does N stand for the number of cyclings or the number of detected photons? On a minor note, please consider adding to Fig. 1b the wave-plate that sets ϕ in the GHZ measurements."

Throughout the manuscript the letter N, as first introduced in line 99 (line 92 in the first submission), refers to the number of detected photons. We also added this information to the caption of Figure 2 in order to avoid misunderstandings.

We added the wave-plate in Fig. 1b.

Referee: "5) Line 47: An explicit citation to an SPDC system (e.g., Ref 3) can be provided at the end of the paragraph."

We followed the recommendation of the referee.

Referee: "6) Line 79: The sentence ("The atom is hence in a superposition. . .") is not strictly correct, as the atom is entangled with a photon."

We corrected our somewhat sloppy formulation and now write: "This process can be written as $|2, 0\rangle \rightarrow (|1, 1\rangle |R\rangle - |1, -1\rangle |L\rangle) / \sqrt{2}$, where $|R/L\rangle$ denotes right/left circular polarization of the photon and $\{|1, 1\rangle, |1, -1\rangle\}$ serves as our atomic qubit basis."

Referee: "7) Line 148: It can be helpful for the nonexperts to add a short comment on the implications of the measured values for the stabilizers."

We added a short explanation regarding the measured values of the stabilizers: "We find an average of $\langle S_1 \rangle = 96.13(9)\%$ and $\langle S_k \rangle = 92(1)\%$ for $k \geq 2$, indicating a large overlap of the generated state with the target linear cluster state."

RESPONSE TO REFEREE #2

We thank the referee for his/her report. We appreciate that the referee considers that “the experiment is an important breakthrough”. His/her remarks will be helpful to provide information to specialist as well as non-specialist readers.

Here is our point-to-point reply:

Referee: “The generation rate for GHZ states is reported as being faster than all the other recent implementations. In particular, in Figure 4, the coincidences rate versus photon-number of the GHZ protocol is compared with the ones of SPDC, Rydberg, and quantum-dot (QD) platforms. However, the reported results for QD platforms concern the generation of cluster states and not of GHZ states. As the generation rate of clusters and GHZ states in the manuscript are different, it would be fair to compare the coincidence rate of the QD experiments with rate of cluster states generation.”

For readability purposes we initially chose to only show a single data set and put the focus on the photon number scaling, which in our view contains the crucial information. However, in light of the referee’s comment, we agree that showing the rate for both GHZ and cluster states provides a more complete picture. We thus updated Figure 4 accordingly.

Referee: “Also, quantum computing requires to go beyond linear cluster, by producing $n \times m$ grid cluster states. In the abstract, the authors predict modular extension via multiple atoms in a single cavity or by coupling several cavities, but given the cited articles (where 2 atoms in a cavity or 2 cavities are coupled) it would be more appropriate to promise $2 \times m$ - i. e. two dimensional - cluster states, which would be in any case an unprecedented result for single-photon cluster states.”

Figure 1a is a long-term vision, there will be intermediate near term steps before. Clearly, using two cavity systems limits to the generation of $2 \times m$ cluster states, and going beyond would require to build additional setups which is technically demanding but otherwise straightforward. On a conceptual level, though, it is easy to argue that the demonstrated two-qubit gate (Ref. 16) is sufficient for the envisioned multi-qubit architecture as only two-qubit gates between neighboring nodes are required.

Similarly, the second strategy of using multiple atoms in one cavity is a priori not limited to $2 \times m$ cluster states in the near future: we are currently implementing optical tweezers to be able to work with more than two atoms.

In our opinion, the potential to scale up to an array of emitters has important implications for photonic quantum computation, as also pointed out by Referee #1, and should not go unmentioned. We have slightly modified the outlook to better explain our next steps.

Referee: “On the same line: the coincidence rates scaling is driven by the overall generation and detection efficiency and in Figure 4 the light blue line indicates the rate corrected by the

detection efficiency. Is there any room for fast improvement? i.e., the reported detection efficiencies (free-space-to-fiber couplings (94% twice), propagation through optical fiber (97%), free-space optics (90%) and detector efficiency (90%)) are already at the state of the art or they can be easily improved?"

The stated efficiencies cannot be easily improved. They are already optimized to above 90%, some near 100%, which does not leave a lot of room for substantial improvements. However, there are two components in the setup which offer the largest potential for improvement. First, a fiber-to-fiber coupling (3-5% losses) could be replaced by a fiber splice. Second, one of the free-space-to-fiber couplings could be removed by rebuilding parts of the setup. These two steps alone would improve the overall efficiency from 43% to about 50%. In our case, however, the required changes would come at the costs of flexibility and convenience, both crucial for demonstration experiments.

Referee: "Also, intrinsic generation efficiency can of course be improved via higher-quality mirrors and a smaller cavity mode volume, but this would require to build a new experimental setup. Those are parameters that probably have already been optimized according to other requirements, is this really a viable solution or it would degrade other figures of merit?"

The referee's implication that a good trade-off between certain figures of merit has to be found, is indeed a very good remark. One might naively think that increasing the cavity finesse is beneficial as it would increase the optical cooperativity. However, at the same time the escape efficiency might decrease due to an increased number of roundtrips and constant parasitic losses. Improving the generation efficiency therefore boils down to finding a compromise between the cooperativity and the escape efficiency.

Referee: "A minor point: the entangling procedure by repeating step 1-4 of the protocol is not easy to follow if the protocol proposed by Lindner and Rudolph in reference 6 is not already known to the reader. I suggest to give the explicit expression of the atom-photons states for at least two cycles in the Methods section."

We followed the suggestion of the referee and added expressions after one and two cycles of the protocol in the Methods.

RESPONSE TO REFEREE #3

We thank the referee for his/her report and valuable input. We are glad that our work is seen as a “major experimental achievement” of “a long-standing challenge in the field.” We appreciate to see that the referee finds that our “achievements warrant publication of the work in Nature”.

Here is our point-to-point reply:

Referee: “The weakest point of the work is potentially that it is not entirely clear what the next step of this research direction would be, i.e. what new opportunities does this work open up? In the introduction the authors state that the work “removes a long-standing obstacle towards scalable measurement-based quantum computation and communication” which is probably somewhat overstated or maybe not sufficiently precisely elaborated in the text. The work demonstrates a way of creating 1D entanglement (which is not a sufficient resource for the mentioned application areas) and it is only hinted how to scale up to more advanced resource states by reflecting photons off the quantum nodes (Fig. 1a).”

We understand that 1D cluster states have limited applications, but believe that extending to 2D has now become quite realistic.

In this context, we agree that the last sentence of the abstract is perhaps not sufficiently clearly formulated, as it might give the false impression that the reported experiment provides the necessary ingredients for measurement-based quantum computation (MBQC). The motivation for the sentence was to stress the importance of generating photonic entanglement efficiently, as this has been one of the major obstacles for MBQC. We now realized that our sentence might have been too enthusiastic and therefore changed it to convey this message more accurately: “Having surpassed previous limitations of probabilistic schemes for photonic entanglement generation, our results open up a new path towards scalable measurement-based quantum computation and communication.”

Given the referee’s remark about the next possible research steps and a similar comment from Referee #2, we changed the outlook of the main text and now provide more details.

Referee: “Another approach would be to de-multiplex the 1D train of single photons and use quantum interference to synthesize more advanced states. However, this would require a high degree of indistinguishability of the emitted photons, which is not measured in the present experiment. Ideally the indistinguishability would be explicitly measured, since this is a key property of an entanglement source. I suggest the authors elaborate this point and give more precise indications of how to scale up the system and what potential applications that would be following.”

The possibility to de-multiplex the 1D train of single photons is an interesting topic we have not addressed in the manuscript. Moreover, we did not explicitly measure the single-photon indistinguishability as this was extensively researched in a previous experiment (Ref. 30), where we showed that we can control the temporal wavefunction of the emitted photon with

a very high degree of precision. We argue that controlling the temporal mode is equivalent to testing the indistinguishability by e.g. two-photon interference. We take the comment of the referee as an indication that the topic might be of interest for the reader, which is why we decided to explicitly mention it in the main text now: “A vacuum-stimulated Raman adiabatic passage (vSTIRAP) enables the generation of photons with high indistinguishability stemming from accurate control over their temporal wavefunction.”

This being said, we note that in the proposed architecture (Fig. 1a) indistinguishability is not necessary for performing entangling operations, as our scheme relies on entangling gates between the emitters (not two-photon interference).

Referee: “Benchmarking with other platforms: the experiment is impressive and certainly the front-runner, which is clear from Fig. 4. However, from reading the manuscript one may get the incorrect impression that the single-atom approach has fundamental advantages compared to other emitters, which I don't think is appropriate. I think a more precise distinction would be between deterministic (Rydberg-based, quantum dots, SiV, or even superconducting qubits) and non-deterministic sources (SPDC). I would therefore suggest a more balanced description of the various deterministic sources including pros and cons. On a related note, I think the reference list is somewhat narrow and could be extended to better reflect the potential of the other deterministic platforms as well.”

We did not intend to give the impression that the neutral-atom platform is fundamentally better suited for photonic entanglement generation. We do understand that other platforms including those mentioned by the Referee can in principle run similar protocols and are viable candidates, too.

We agree that the distinction between deterministic and probabilistic sources is important, as we argue in the introduction. This being said, we believe that a balanced comparison of performance has to be made between all platforms. Only comparing to deterministic sources would not correctly reflect the state-of-the-art, considering that probabilistic SPDC systems were hitherto the clear leader in the field.

The Referee's request for a more extended description of the pros and cons of other deterministic platforms is hard to meet given the length limit of a concise research paper like ours. The reason is that different platforms experience different limitations which cannot be summarized in just a few sentences. This includes our own system, where the big step from two entangled photons (Wilk et al., *Science* **317**, 488 (2007)) to 12/14-photon cluster/GHZ states required a plethora of new insights and novel implementations. Addressing the potential of other platforms beyond objective criteria such as the published coincidence rates referenced in Figure 4 in a way that is useful for the reader requires, in our opinion, a broader review.

Concerning the list of references we note that we cited the, to the best of our knowledge, most advanced deterministic platforms demonstrating multi-photon entanglement. We have, nonetheless, expanded the reference list by three recent publications on efficient single-photon sources based on quantum dots (Senellart et al., *Nat. Nanotechnol.* **12**, 1026–1039 (2017), Tomm et al., *Nat. Nanotechnol.* **16**, 399–403, (2021)) and SiV centers (Knall et al., *ArXiv preprint* 2201.02731 (2022)), all of which reflect the progress being made towards efficient photonic entanglement generation.

Referee: "The sentence "By virtue of this feature, so far unique to the atomic CQED platform.." is not entirely clear. Why is this unique to atomic systems as opposed to other deterministic sources?"

We agree with the referee's comment that the described mechanism also applies to other deterministic sources, at least in principle. However, to this day, performing quantum gates between remote qubits in a network architecture has only been demonstrated experimentally with neutral atoms in optical cavities. We believe that by saying "so far" we implicitly state the potential for other platforms to achieve the same in the future.

Referee: "It would be interesting to have further details on the various contributions to infidelities in the experiment for instance by breaking down the protocol in the various steps and explain the error contributions (e.g. in Supplementary Information). That would help understand what are the critical steps for improving even further in the future."

The same point was raised by the Referee #1. In short, breaking down those very small errors is hard experimentally given the numerous potential sources. However, we added a new figure in the Methods that we see as the smoking gun that the main limitation in the fidelity is the ν STIRAP process due to the limited cooperativity. However, the current theoretical models fail to describe our results accurately and hence it is not yet possible to make predictions about the necessary improvements. The cooperativity being also a limitation for the efficiency, it is clear that it needs to be further optimized.

Referee: "Also it would be instructive with a little more experimental details. How many actual coincidence events (14 fold) are detected and what was the experimental measurement time? What kind of post-selection is implemented and further elaboration of the role of the experimental duty-cycle could be given. What are the specs of the Z/X switch and how is the correlation detection carried out? In general an elaborated Supp. Info with such experimental details would be beneficial. "

We added some details in the manuscript (main text and Methods).

Reviewer Reports on the First Revision:

Referees' comments:

Referee #1 (Remarks to the Author):

The authors have satisfactorily addressed most of my concerns in the revised manuscript and reply letter.

The text now states that the finite cooperativity ($C \sim 1.5$) limits the generated state fidelity (fidelity reduction of $S = 1\% \sim 2\%$ per photon). It would be great to have a simple expression connecting these two crucial figures. Perhaps a rough estimation that includes the detuning from the unwanted emission paths [e.g. $S = (1/C) * (\Gamma/\Delta)$ or so]. This could have helped the reader better understand the path to future improvement, which is one issue raised by all three referees. I do understand from the reply letter that current theoretical models do not quantitatively capture the fidelity reduction, but I still wish there was a qualitative estimation.

As before, I strongly support the publication of the paper in Nature.

Referee #2 (Remarks to the Author):

The authors answer and clarifications in the text are satisfactory, so I recommend the manuscript for publication

Referee #3 (Remarks to the Author):

The authors have responded to all comments from the reviewers in a satisfactory manner. I recommend publication of the manuscript

Author Rebuttals to First Revision:**RESPONSE TO REFEREE #1**

We would like to thank the referee for his/her recommendation supporting publication in *Nature* and will try to clarify the remaining point.

Referee: "The text now states that the finite cooperativity ($C \approx 1.5$) limits the generated state fidelity (fidelity reduction of $S=1\%-2\%$ per photon). It would be great to have a simple expression connecting these two crucial figures. Perhaps a rough estimation that includes the detuning from the unwanted emission paths [e.g. $S=(1/C)(\text{Gamma}/\Delta)$ or so]. This could have helped the reader better understand the path to future improvement, which is one issue raised by all three referees. I do understand from the reply letter that current theoretical models do not quantitatively capture the fidelity reduction, but I still wish there was a qualitative estimation."*

We agree with the referee that a simple expression connecting the cooperativity with the fidelity would be helpful. Qualitatively it is clear that a higher cooperativity also leads to a higher fidelity. However, we have neither been able to develop nor find a model in the literature that quantitatively describes our experimental results. The topic therefore remains the subject of future research.

RESPONSE TO REFEREE #2

We are glad that we were able to resolve all comments brought forward by the referee and are grateful for his/her recommendation supporting publication in *Nature*.

RESPONSE TO REFEREE #3

We are glad that we were able to resolve all comments brought forward by the referee and are grateful for his/her recommendation supporting publication in *Nature*.